# Association of Abnormal Iron Status with the Occurrence and Prognosis of Peritoneal Dialysis-Related Peritonitis: A Longitudinal Data-Based 10-Year Retrospective Study

**DOI:** 10.3390/nu14081613

**Published:** 2022-04-13

**Authors:** Xiangwen Diao, Zhiwei Zheng, Chunyan Yi, Peiyi Cao, Hongjian Ye, Ruihua Liu, Jianxiong Lin, Wei Chen, Haiping Mao, Fengxian Huang, Xiao Yang

**Affiliations:** 1Department of Nephrology, The First Affiliated Hospital, Sun Yat-sen University, No. 58 Zhongshan Er Road, Guangzhou 510080, China; diaoxw@mail2.sysu.edu.cn (X.D.); yichy@mail.sysu.edu.cn (C.Y.); jesse-cao@163.com (P.C.); yehongj@mail.sysu.edu.cn (H.Y.); liurh5@mail2.sysu.edu.cn (R.L.); ljianx@mail.sysu.edu.cn (J.L.); chenwei99@mail.sysu.edu.cn (W.C.); maohp@mail.sysu.edu.cn (H.M.); huangfx@mail.sysu.edu.cn (F.H.); 2Guangzhou Center for Disease Control and Prevention, Guangzhou 510440, China; zhengzhw5@mail2.sysu.edu.cn; 3Key Laboratory of Nephrology, Ministry of Health and Guangdong Province, Guangzhou 510080, China

**Keywords:** peritoneal dialysis, peritonitis, iron status, functional iron deficiency, high iron status, prognosis

## Abstract

This retrospective study investigated the effect of iron status on peritonitis by analyzing longitudinal iron parameters in peritoneal dialysis (PD) patients. Patients who received PD at our center from 1 January 2006 to 31 December 2015 were included and followed up until 31 December 2017. According to the joint quartiles of baseline transferrin saturation and ferritin, iron status was categorized as reference iron status (RIS), absolute iron deficiency (AID), functional iron deficiency (FID), and high iron status (HIS). Generalized estimating equations and Cox regression models with time-dependent covariates were used. A total of 1258 PD patients were included; 752 (59.8%) were male, with a mean (±standard deviation) age of 47.4 (±14.9) years. During a median follow-up period of 35.5 (interquartile range, 18.4–60.0) months, 450 (34.3%) patients had 650 episodes of peritonitis. By analyzing longitudinal data, patients with AID were independently positively associated with the occurrence (adjusted odds ratio (AOR) = 1.45) and treatment failure of peritonitis (adjusted hazard ratio (AHR) = 1.85). Patients with HIS were positively associated with the treatment failure of peritonitis (AHR = 2.70). Longitudinal AID and HIS were associated with the episodes and poor prognosis of peritonitis. Active clinical monitoring and correction of iron imbalance in patients with PD are needed.

## 1. Introduction

Iron metabolism is closely related to human infectious diseases. On the one hand, the bactericidal function of neutrophils and the proliferation of lymphocytes depend on the presence of iron; on the other hand, iron is essential for almost all infectious microorganisms’ growth, especially Siderophilic bacteria, which compete with transferrin to bind unbound iron [1]. In recent years, intravenous iron therapy for renal anemia has been rapidly promoted. However, for peritoneal dialysis (PD) patients, oral iron is still the mainly adopted form of iron supplementation [2,3]. Regardless of the type of iron supplementation, it can be observed clinically that a certain proportion of PD patients still have increased or decreased iron levels [4,5]. Whether the abnormal iron status can lead to increased infection rate and affect the treatment outcome of infection is still controversial [6].

Though the incidence of PD-associated peritonitis has decreased due to technological advances, it remains a serious complication that threatens the prognosis of PD patients, including technique survival [7]. Iron plays an important pathophysiological role in both PD patients and peritonitis-causing pathogens. A recent dialysate proteomics study demonstrated that the saturation ratio of transferrin in the dialysate was higher than that in the plasma; transferrin in the dialysate could provide a direct source of iron for bacterial growth [8]. However, the relationship between serum iron status and PD-related peritonitis is still unknown. Therefore, in this study, we investigated iron status and its effects on episodes and prognosis of peritonitis in PD patients by analyzing longitudinal iron parameters.

## 2. Materials and Methods

### 2.1. Study Population

Patients who underwent PD at our center from 1 January 2006 to 31 December 2015 were included in this study and were followed up until 31 December 2017. The exclusion criteria of the study were as follows: (1) age < 18 years, transferred from long-term hemodialysis or renal transplant failure; (2) discontinued PD or transfer to another center within first 3 months; (3) having a history of malignancy; and (4) lack of relevant data. Patient-specific inclusion and exclusion processes are shown in Figure 1. All patients used standard lactate–glucose peritoneal dialysate (1.5%, 2.5%, or 4.25% dextrose; Baxter, Guangzhou, China). The majority of patients in our center had adopted oral iron supplementation, including Iron Polysaccharide Complex Capsules 0.15 qd-bid, Ferrous Succinate Tablets 0.1–0.2 bid, etc. The study was approved by the clinical research ethics committee of the First Affiliated Hospital of Sun Yat-sen University (Ethical review No. [2016]215), and was performed in accordance with the Declaration of Helsinki.

### 2.2. Data Collection

Demographic information, laboratory parameters, and medications were collected at baseline. The baseline and longitudinal data of serum iron, transferrin saturation (TSAT), ferritin, hemoglobin (Hb), albumin (Alb), neutrophil/lymphocyte ratio, and high-sensitivity C-reactive protein (hs-CRP) were included, but the data from 4 weeks before the diagnosis of peritonitis to 4 weeks after the end of peritonitis treatment were excluded. The follow-up process was divided into 20 phases. The baseline phase was the first month to the sixth month after PD initiation. After that, every 6 months was defined as a follow-up phase, and the last one was more than 10 years. If an index, for example ferritin, had multiple measured values within a follow-up phase, the average value of that index was calculated as the value of that index in the period. Based on [9,10], iron status was classified by using baseline iron indices as follows: reference iron status (RIS), 2nd and 3rd quartiles of TSAT and ferritin; absolute iron deficiency (AID), 1st quartiles of TSAT and ferritin; functional iron deficiency (FID), 1st TSAT quartile with 3rd and 4th quartiles of ferritin; and high iron status (HIS), 4th quartiles of TSAT and ferritin. The iron status of patients in each follow-up phase was re-classified according to their iron index results in each period.

### 2.3. Outcomes

Peritonitis was defined as the presence of at least two indices: abdominal pain or pressure; peritoneal effluent leukocytes ≥ 100 cells/mL, with at least 50% neutrophils; and a positive dialysate microbiological culture [11]. Treatment failure for peritonitis was defined as temporary or permanent termination of peritoneal dialysis, and death during peritonitis; peritonitis-related mortality was defined as death with active peritonitis or within 4 weeks of a peritonitis episode, or any death during hospitalization for peritonitis [12].

### 2.4. Statistical Analyses

Mean ± standard deviation (SD) or median (interquartile range (IQR)) was used to describe numerical variables; frequency and percentage were used for subtype variables. Variance analysis or Kruskal–Wallis H test was applied to compare differences in numerical variables between different iron status at baseline. Rates were compared by chi-squared test. The effect of baseline iron status on the frequency of peritonitis and the first episode of peritonitis were explored by negative binomial regression models and Cox regression analyses, respectively. Generalized estimating equations (GEE) were used to explore the relationship between longitudinal iron status and the occurrence of peritonitis. The effect of longitudinal iron status on the prognosis of peritonitis were probed by Cox regression with time-dependent covariates. All statistical analyses were performed using R 3.6.1 software.

## 3. Results

### 3.1. Clinical Characteristics between Different Iron Groups at Baseline

Figure 1 depicts the entire patient screening process. During a median follow-up period of 45.1 (26.8–70.1) months, a total of 1483 patients with PD were eligible, and 9687 longitudinal (including baseline) iron indices records were collected. Based on the baseline TSAT and Ferritin quartiles of these 1483 patients (see Figure 2), 2650 iron index records were classified as RIS, 593 as AID, 473 as FID, and 929 as HIS. The remaining 5042 records did not fall into any of these four categories, and were thus excluded from subsequent analyses. Finally, a total of 1258 PD patients were included, of which 752 (59.8%) were male and 309 (24.6%) were diabetic, with a mean (±standard deviation) age of 47.4 (±14.9) years. Among the 1258 included PD patients, 1245 patients were being treated with continuous ambulatory PD (CAPD) and 13 patients were being treated with automated PD (APD). A total of 340 patients performed PD with assistance, of which 324 were assisted by family members and 16 by others. Table 1 compares the clinical characteristics of the four groups at baseline. Compared with other groups, the AID group had the lowest male percentage and levels of BMI and uric acid (*p* < 0.05, Table 1); the FID group had the highest level of hs-CRP and neutrophil/lymphocyte ratio (*p* < 0.01) as well as the lowest level of Hb; and the HIS group had the highest male percentage and levels of uric acid, as well as the lowest use ratio of erythropoietin stimulating agents (ESA) (*p* < 0.05).

### 3.2. Baseline Iron Status Was Not Associated with the Occurrence of Peritonitis

During a median follow-up period of 35.5 (IQR, 18.4–60.0) months, a total of 650 peritonitis events occurred in 450 (35.8%) patients. Among them, 239 (53.1%) cases had one episode of peritonitis, 78 (17.3%) patients had two episodes, and 62 patients (13.8%) had at least three episodes. The overall incidence of peritonitis was 0.15 per patient−year (95% confidence interval (CI): 0.14–0.16). Neither univariate and multivariate negative binomial regression nor Cox regression analysis found any association between baseline iron status and the occurrence of peritonitis (Appendix A).

### 3.3. Relationship between Longitudinal Iron Status and the Occurrence of Peritonitis

Data on iron biomarkers, Hb, and Hs-CRP during the follow-up period are shown in Appendix A. The distribution of iron status records at each follow-up phase is shown in Figure 3 and Appendix A. After PD and iron supplementation, the proportions of AID and FID groups decreased to a certain extent, especially FID, and the proportions of RIS increased; meanwhile, the proportion of HIS stabilized between 7% to 12%. Univariate GEE analysis showed that compared with RIS, AID (odds ratio (OR) = 1.72, 95% CI: 1.22–2.42, *p* = 0.002, Table 2) and FID (OR = 1.93, 95% CI: 1.43–2.60, *p* < 0.001) were positively associated with the occurrence of peritonitis. After adjusting for age, gender, Charlson comorbidity index, PD vintage, Alb, Hb, Hs-CRP, and neutrophil/lymphocyte ratio, AID (adjusted OR = 1.45, 95% CI: 1.06–2.00, *p* = 0.02, Table 2) was independently positively associated with the occurrence of peritonitis.

### 3.4. Relationship between Longitudinal Iron Status and the Prognosis of Peritonitis

Among the 650 episodes of peritonitis, 568 episodes were cured (87.4%), 82 episodes suffered treatment failures (12.6%), and 24 patients died from peritonitis (3.69%). Univariate Cox regression analyses with time-dependent covariates showed that AID (hazard ratio (HR) = 2.16, 95% CI: 1.17–3.98, *p* = 0.01, Table 3), FID (HR = 2.14, 95% CI: 1.11–4.14, *p* = 0.02), and HIS (HR = 2.33, 95% CI: 1.28–4.24, *p* = 0.005) were risk factors for treatment failure. After adjusting for age, gender, diabetes, Alb, Hb, Hs-CRP, and neutrophil/lymphocyte ratio, AID (adjusted HR = 1.85, 95% CI: 1.01–3.39, *p* = 0.04, Table 3) and HIS (adjusted HR = 2.70, 95% CI: 1.39–5.25, *p* = 0.003) were found to be independent risk factors for treatment failure.

## 4. Discussion

In this study, we explored the relationship between iron status and episodes as well as prognosis of peritonitis in PD patients. Baseline iron status was not associated with the incidence of peritonitis. By analyzing longitudinal data, we found that compared to those with RIS, patients with AID were independently positively associated with the occurrence of peritonitis; AID and HIS were independent risk factors for the treatment failure of peritonitis.

AID and FID are two common types of iron deficiency in patients with chronic kidney disease (CKD), especially dialysis. The former manifests as reduced levels of both circulating and stored iron, while the latter only has a deficiency of circulating iron [6]. AID was traditionally defined as TSAT < 20% and ferritin < 100 ng/mL, while FID was TSAT < 20% and ferritin > 100 ng/mL [6,13,14,15]. Two recent clinical studies focused on iron and clinical outcomes in CKD patients used the quartile of TSAT and Ferritin to define iron status [9,10]. The quartile (0%, 25%, 50%, 75%, 100%) of TSAT (%) in the cohort of Cho et al. [9] was 0.4, 16.1, 21.6, 28.0, 99.6 and the quartile of ferritin (ng/mL) was 0.4, 55.0, 106, 205, 4941. The quartile of TSAT (%) in the study of Mehta et al. [10] was 1.3, 17.3, 22.1, 28.0, 87.1 and the quartile of ferritin (ng/mL) was 4.9, 82.5, 158, 284, 3769. We wanted to establish four non-overlapping iron groups using iron thresholds specific to our PD study cohort. The quartile of TSAT (%) in our study population was 0.6, 12.3, 18.3, 27.5, 99.7 and the quartile of ferritin (ng/mL) was 4.0, 64.0, 147, 282, 2173, which were close to the threshold values of the previous two studies.

Our study revealed that longitudinal AID was independently associated with the occurrence and treatment failure of peritonitis. A systematic review including four prospective cohort studies and two retrospective studies found that iron deficiency or iron deficiency anemia was associated with a higher occurrence of different types of infections in different patient populations [16]. A study focused on HD patients showed that low TSAT (<20%) was independently associated with infection-related deaths [17]. However, these studies did not adjust for possible confounders of the association between iron status and infection. In this study, we took care to adjust the models for confounding factors such as inflammation, anemia, and serum albumin, and demonstrated that low iron was associated with PD-related peritonitis. Iron plays an important role in human immunity. First, iron is essential for the normal proliferation of intestinal epithelial, which constitutes a physical barrier to infection [18]. Second, free iron contributes to the formation of reactive oxygen species, which is essential for the maintenance of phagocytic function [19]. Studies have shown that iron deficiency can impair both cellular and humoral immunity, particularly T-cell immunity [20,21]. Iron deficiency also affects tissue energy metabolism and muscle function, resulting in damage to cardiovascular and other organ functions [22,23]. Also, recent studies have found that pathogens can mobilize iron even in iron-deficient states. Many pathogenic bacteria, including Staphylococcus aureus, Streptococcus, Salmonella, and Escherichia coli, produce hemolysin to destroy erythrocyte for iron [24,25]. Studies have shown that when iron is deficient, fungi activate a series of strategies to capture exogenous iron, mobilize iron stored in cells, and regulate their own metabolism to conserve iron [26]. These data may explain why low iron is associated with PD-related peritonitis.

Another type of iron deficiency, FID, was not found to be independently associated with the occurrence and treatment failure of peritonitis, while Alb and neutrophil/lymphocyte ratio were identified as associated factors. These results indicated that the effect of iron deficiency on infection risk was confounded by factors such as inflammation and malnutrition. The inflammatory effects of peritonitis can lead to FID. Inflammation can increase the expression of hepatic hepcidin through the IL6-JAK2-STAT3 signaling pathway, resulting in a decrease in the amount and function of the iron exporter ferroportin in cells; iron cannot be transported out of the cell, and plasma iron concentration decreases, limiting the availability of iron by extracellular microorganisms [27]. On the other hand, there is a price to pay for this defense mechanism. In addition to the hematopoietic system, iron is also indispensable for the body’s immune system. Iron deficiency impairs both innate and adaptive immune function [19,20,21]. Therefore, patients with FID also appear to be prone to peritonitis, forming a vicious circle.

Previous studies have demonstrated that high serum ferritin (usually defined as >500 or 1000 ng/mL) is associated with infection in patients with predialysis-CKD and HD [28,29,30,31]. Retrospective data suggests that intensive intravenous iron administration might be associated with increased risk of mortality and infections [32]. However, recent studies have not observed that relationship. A systematic review concluded that high doses of intravenous iron in dialysis patients were not associated with an increased risk of infection [33]. A recently randomized controlled trial PIVOTAL revealed that the high-dose and low-dose intravenous iron groups had the same infection rates [34]. The relationship between high iron and infection in PD patients was also not observed. Vychytil et al. treated TSAT < 20% PD patients with 200 mg/month intravenous sucrose iron, and TSAT > 20% patients with 100 mg/month for 6 months, and found that the incidence of catheter infections and peritonitis was the same [35]. Allen et al. conducted a retrospective study in the PD population and did not find an increase in risk of peritonitis between patients receiving and not receiving intravenous iron [36]. We also failed to show an association between high iron and peritonitis by using TSAT and ferritin to specifically define high iron status. However, when infection occurs, sufficient iron promotes the growth of pathogens [37]. Excessive iron can impair monocyte and neutrophil function [38,39]. Therefore, the latest guidelines insist that iron supplementation during active infection is not recommended [6]. The finding of our study that HIS was associated with treatment failure in patients with peritonitis also supports this view.

We found that AID, HIS, and decreased levels of albumin were independently associated with the treatment failure of peritonitis, and FID also showed a trend related with the treatment failure of peritonitis. This reflects the collaborated effect of infection risk factors such as abnormal iron metabolism, malnutrition, and inflammation that contribute to the persistence of peritonitis episodes. Hypoalbuminemia is the most commonly used surrogate of Protein-Energy Wasting (PEW) and malnutrition in dialysis patients [40]. Low serum albumin level is associated with persistent systemic inflammation and is also a marker of inflammation [40]. Inflammation plays a dominant role in the pathophysiology of PEW and malnutrition [41]. Inflammation leads to malnutrition and iron metabolism disorder in patients; malnutrition and iron metabolism disorder lead to immune function deficiency that promotes the occurrence of infection, which in turn aggravates inflammation. The interaction and synergy of these risk factors ultimately lead to unresolved peritonitis in patients.

Both transferrin and ferritin are acute phase reactants with a relatively short half-life [42,43]. Through intervention, the baseline iron status may change its composition ratio over prolonged follow-up periods, and affect the association between baseline iron status and peritonitis. We found that longitudinal low iron status, not baseline iron status, was associated with the episodes and poor prognosis of peritonitis in PD patients. These results suggest that persistent iron imbalance might be more liable to peritonitis. 

The strengths of this study include a relatively large PD sample size, benefitting from the fact that our center is the largest PD center in southern China. We analyzed not only baseline data, but also longitudinal iron status; in addition, long-term, regular follow-up sessions were conducted for most patients. However, there are limitations to this study. First, this study is a single-center study; center-specific effects cannot be excluded. Second, due to the retrospective study design, some potentially important confounding factors such as hepcidin could not be obtained in this study, as well as more sensitive indicators reflecting iron stores, such as reticulocyte hemoglobin and the percentage of hypochromic cells. Third, data on the sources of infection and causative organisms were not provided. Finally, we did not include iron supplement for statistical analysis because of missing data at follow up. 

In summary, we revealed that AID was strongly associated with the occurrence of PD-related peritonitis, and AID and HIS were risks factor for the poor prognosis of peritonitis. This study utilized clinical real-world data to emphasize the importance of iron balance for the prevention and treatment of peritonitis in PD patients. Future prospective clinical studies using more accurate monitoring of iron status are needed to provide further evidence for standardized clinical iron supplementation in PD patients.

## Figures and Tables

**Figure 1 nutrients-14-01613-f001:**
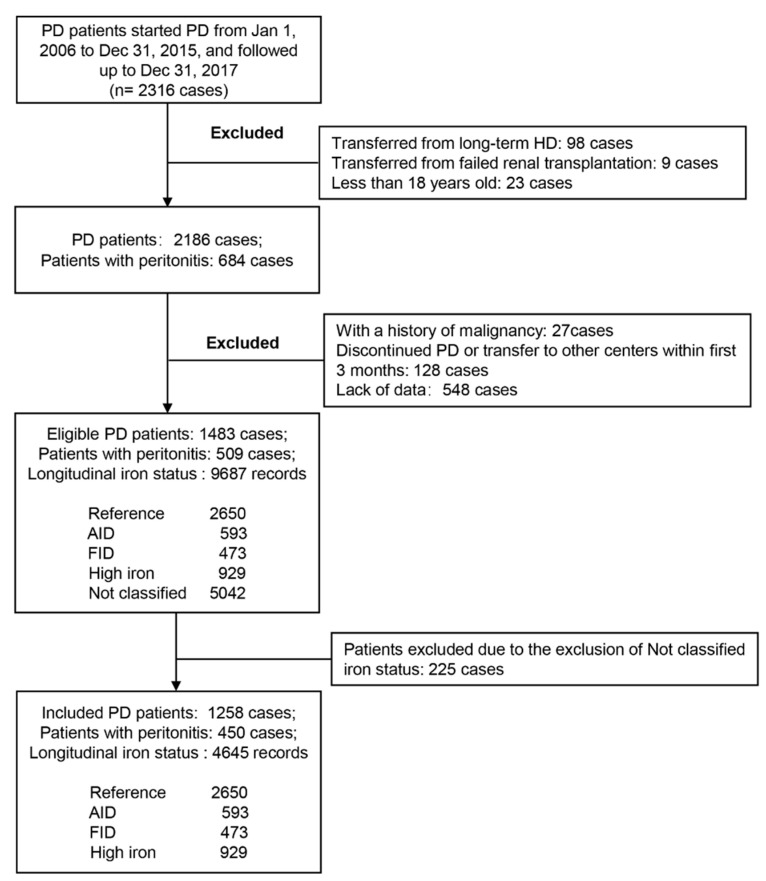
Flowchart of the subject screening process.

**Figure 2 nutrients-14-01613-f002:**
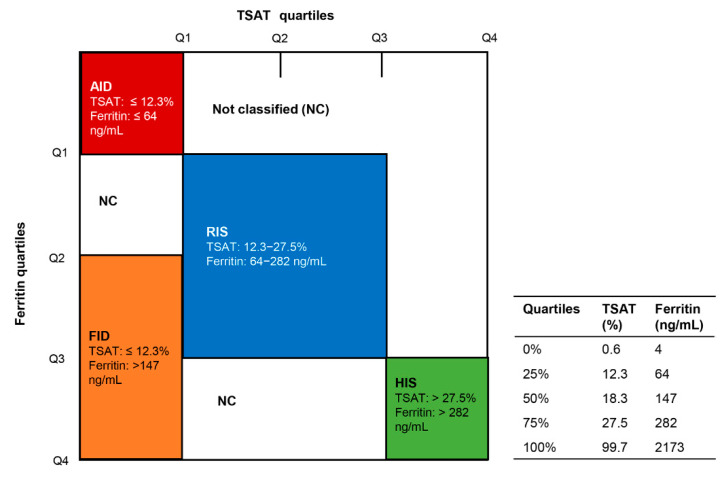
Definition of four iron statuses by joint TSAT and ferritin quartiles.

**Figure 3 nutrients-14-01613-f003:**
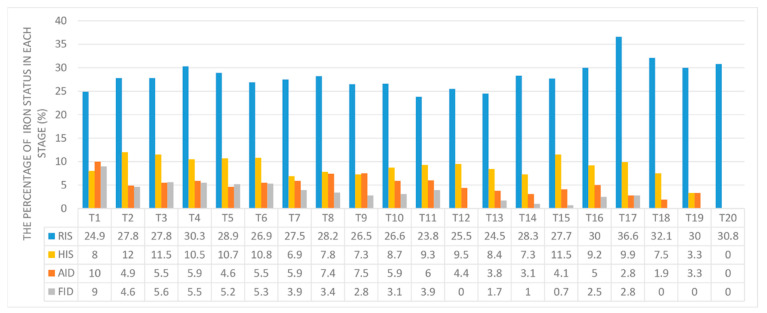
The distribution of iron status records at each follow-up phase.

**Table 1 nutrients-14-01613-t001:** Baseline basic characteristics of subjects.

Variable	Baseline Iron Status				*p* Value
	RIS (*n* = 369)	AID (*n* = 148)	FID (*n* = 134)	HIS (*n* = 118)	
**Demographics**					
Male, *n* (%)	240 (65.0)	45 (30.4)	92 (68.7)	90 (76.3)	**<0.001**
Age (years)	48.1 ± 14.8	46.8 ± 15.9	48.0 ± 15.6	44.1 ± 15.4	0.07
BMI (kg/m^2^)	21.8 ± 3.01	21.1 ± 3.35	21.6 ± 2.93	21.5 ± 2.98	**0.04**
Diabetes, *n* (%)	86 (23.3)	32 (21.6)	33 (24.6)	29 (24.6)	0.93
Assisted PD	101 (27.4)	39 (26.4)	40 (29.9)	22 (18.6)	0.19
CCI	3.53 ± 1.70	3.48 ± 1.846	3.56 ± 1.85	3.46 ± 1.81	0.76
Primary kidney disease					0.61
Chronic glomerulonephritis, *n* (%)	225 (61.0)	91 (61.5)	68 (50.7)	71 (60.2)	
Diabetic nephropathy, *n* (%)	74 (20.1)	28 (18.9)	30 (22.4)	27 (22.9)	
Hypertensive nephropathy, *n* (%)	29 (7.9)	11 (7.4)	15 (11.2)	10 (8.5)	
Other, *n* (%)	41 (11.1)	18 (12.2)	21 (15.7)	10 (8.5)	
**Laboratory parameter**					
Hemoglobin (g/dL)	11.3 ± 1.54	10.8 ± 1.83	10.2 ± 1.98	10.6 ± 2.03	**<0.001**
Hs-CRP (mg/L)	1.73 (0.68–5.64)	1.47 (0.53–3.91)	3.83 (1.03–10.8)	1.62 (0.59–3.61)	**<0.001**
Neutrophil/lymphocyte ratio	2.63 (2.13–3.59)	2.74 (2.22–3.80)	3.11 (2.36–4.03)	2.76 (2.09–3.70)	**0.01**
Albumin (g/L)	37.5 ± 4.28	37.7 ± 4.83	36.8 ± 5.65	36.9 ± 4.83	0.36
Serum creatinine (μmol/L)	684 (563–889)	671 (571–784)	735 (558–984)	768 (599–951)	**0.01**
Total cholesterol (mmol/L)	5.00 (4.30–5.80)	5.10 (4.40–5.90)	5.10 (4.20–5.80)	4.95 (4.20–5.70)	0.68
Urea acid (μmol/L)	420 ± 92.6	398 ± 85.2	420 ± 107	435 ± 80.3	**0.005**
Residual GFR (mL/min/1.73 m^2^)	5.62 (4.13–7.15)	4.63 (3.30–6.76)	4.63 (3.87–6.14)	5.48 (4.19–7.67)	**0.003**
Serum iron (μmol/L)	10.7 ± 3.04	5.15 ± 2.25	5.54 ± 2.53	17.2 ± 5.15	**<0.001**
Transferrin saturation (%)	19.0 (15.7–22.9)	7.85 (5.66–10.2)	9.42 (7.42–11.0)	35.3 (30.5–43.0)	**<0.001**
Ferritin (ng/mL)	147 (110–209)	26.5 (15.6–41.9)	266 (199–461)	458 (359–670)	**<0.001**
**Medications**					
ESAs users, *n* (%)	310 (87.3)	141 (96.6)	115 (91.3)	87 (77.0)	**<0.001**
Iron users, *n* (%)	259 (73.0)	113 (77.4)	87 (69.0)	72 (63.7)	0.09

RIS—reference iron status; AID—absolute iron deficiency; LIS—low iron storage; FID—functional iron deficiency; HIS—high iron status; BMI—body mass index; CCI—Charlson’s comorbidity index; Hs-CRP—high-sensitivity C-reactive protein; GFR—glomerular filtration rate; ESA—erythropoietin stimulating agents.

**Table 2 nutrients-14-01613-t002:** Univariate and multivariate GEE analysis of longitudinal iron status and the episodes of peritonitis in PD patients.

Variables	Univariate Model		Multivariate Model	
	OR (95% CI)	*p* Value	AOR (95% CI)	*p* Value
Age	1.02 (1.01, 1.03)	**<0.001**	1.02 (1.00, 1.03)	**0.03**
Male gender	0.80 (0.63, 1.03)	0.08	0.90 (0.69, 1.16)	0.39
Diabetes	1.02 (0.78, 1.33)	0.88	-	-
CCI	1.13 (1.06, 1.21)	**<0.001**	0.92 (0.82, 1.04)	0.18
PD vintage (per 0.5 year)	1.02 (0.99, 1.05)	0.19	0.99 (0.96, 1.02)	0.52
ALB	0.86 (0.84, 0.88)	**<0.001**	0.87 (0.84, 0.89)	**<0.001**
Hb	0.99 (0.98, 0.99)	**<0.001**	1.00 (0.99, 1.01)	0.97
Hs-CRP	1.02 (1.01, 1.03)	**<0.001**	1.00 (0.99, 1.01)	0.36
N/L ratio	1.13 (1.07, 1.19)	**<0.001**	1.06 (1.01, 1.11)	**0.02**
Iron status				
RIS	Ref.		Ref.	
AID	1.72 (1.22, 2.42)	**0.002**	1.45 (1.06, 2.00)	**0.02**
FID	1.93 (1.43, 2.60)	**<0.001**	1.33 (0.92, 1.91)	0.13
HIS	1.22 (0.92, 1.63)	0.17	1.25 (0.91, 1.71)	0.17

GEE—generalized estimating equations; CCI—Charlson’s comorbidity index; ALB—albumin; Hs-CRP—high-sensitivity C-reactive protein; N/L ratio—neutrophil/lymphocyte ratio; LIS—low iron storage; FID—functional iron deficiency; RIS—reference iron status; HCI—high circulating iron; HIS—high iron status.

**Table 3 nutrients-14-01613-t003:** Univariate and multivariate cox regression analysis with time-dependent covariates for longitudinal iron status and the treatment failure of peritonitis.

Variables	Univariate Model		Multivariate Model	
	HR (95% CI)	*p* Value	AHR (95% CI)	*p* Value
Age	1.00 (0.99, 1.02)	0.81	0.99 (0.97, 1.01)	0.26
Male gender	0.77 (0.49, 1.21)	0.26	0.75 (0.46, 1.21)	0.24
Diabetes	1.37 (0.84, 2.23)	0.21	1.61 (0.92, 2.85)	0.10
CCI	1.09 (0.96, 1.23)	0.20	-	-
ALB	0.92 (0.88, 0.96)	**<0.001**	0.92 (0.87, 0.98)	**0.008**
Hb	0.98 (0.97, 0.99)	**0.002**	0.99 (0.98, 1.01)	0.13
Hs-CRP	1.01 (1.00, 1.01)	**0.008**	1.00 (0.99, 1.01)	0.38
N/L ratio	1.05 (0.99, 1.11)	0.07	1.01 (0.94, 1.08)	0.80
Iron status				
RIS	Ref.		Ref.	
AID	2.16 (1.17, 3.98)	**0.01**	1.85 (1.01, 3.39)	**0.04**
FID	2.14 (1.11, 4.14)	**0.02**	1.89 (0.97, 3.69)	0.06
HIS	2.33 (1.28, 4.24)	**0.005**	2.70 (1.39, 5.25)	**0.003**

CCI—Charlson’s comorbidity index; ALB—albumin; Hs-CRP—high-sensitivity C-reactive protein; N/L ratio— neutrophil-lymphocyte ratio; FID—functional iron deficiency; RIS—reference iron status; HCI—high circulating iron; HIS—high iron status.

## Data Availability

The datasets generated during and/or analyzed during the current study are available from the corresponding author upon reasonable request.

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
