# Peer review of "Association of Abnormal Iron Status with the Occurrence and Prognosis of Peritoneal Dialysis-Related Peritonitis: A Longitudinal Data-Based 10-Year Retrospective Study"

_nutrients, 2022, doi:10.3390/nu14081613_

Round 1

Reviewer 1 Report

This study focuses on an important area. As you state, the large patient number is another strength.

I have some specific comments:

1) The fact that this is a retrospective study should be also clarified in the title. Furthermore, please, write in the first line of the abstract " …This retrospective study investigated the effect of ironstatus……"

2) Why was this study submitted 2022, while the observation period ended 2017?

3) Some important information should be added:

a) Which PD fluids have been used in included patients (Low GDP fluids? Conventional PD fluids with acidic pH? How many patients received icodextrin-containing PD fluid?)

b) Were patients exclusively treated with CAPD, or did part of the patients perform APD?

c) Did you use antibiotic prophylaxis (e.g. mupirocin ointment) in included patients?

d) Did some of the patients need assistance for perfoming PD?

If there are differences between groups these parameters (a-d) should be included in the multivariate model. However, if these data are not available this should at least be considered as limitation in the manuscript.

4) How did you consider relapsing peritonitis and reinfection?

5) It would be interesting to know sources of infection of peritonitis (e.g. how many cases were associated with catheter-tunnel infection?) and causative organisms. This could allow better interpretation of your data. If these data are not available this should also be stated as limitation.

6) Table 1: Did iron users receive exclusively oral iron, or did part of the patients receive intravenous supplementation?

Reviewer 2 Report

The authors propose a large study in the patient on peritoneal dialysis. 1483 patients are enrolled with an  average follow-up period of more than 40 months. The authors analyze the relationship between ferric status, patients being grouped into 4 categories, and the occurrence of an episode of peritonitis.  Patients with absolute deficiency are at higher risk of peritonitis and patients with a higher load are likely to be resistant to treatment.

This study on a large population followed over the long term provides elements that can influence the follow-up of the patient on peritoneal dialysis and the management of the treatment of anemia and iron deficiency. The discussion is well constructed around an up-to-date bibliography.

Nevertheless, I have a few remarks and a point to clarify that seems to me to be major:

  • For the nephrologist reader it would be interesting to specify the type of treatment (CPD or APD) and the type of dialysate used (frequency of hypertonic pockets and / or amino acids)
  • HIS patients despite high status (TSAT 35%, Ferritin 458) are iron users without difference with other groups. Does a relationship exists between “over-treatment” in these patients and failure of the relative resistance to the peritonitis treatment?
  • There is no relationship with status at inclusion. In the methods it is not clear how deficit patients are treated and in general if there is a relationship with the treatment (forms, doses). It can be assumed that some patients respond and others do not. In summary in the methods it would be better to detail, during the longitudinal follow-up how patients are classified. It seems that patients are re-classified every 6 months. Is the riskcalculated according to the patient's status at the time of the peritonitis episode?
